# The Intrinsically Disordered N Terminus in Atg12 from Yeast Is Necessary for the Functional Structure of the Protein

**DOI:** 10.3390/ijms242015036

**Published:** 2023-10-10

**Authors:** Hana Popelka, Vikramjit Lahiri, Wayne D. Hawkins, Felipe da Veiga Leprevost, Alexey I. Nesvizhskii, Daniel J. Klionsky

**Affiliations:** 1Life Sciences Institute, University of Michigan, Ann Arbor, MI 48109, USA; vlahiri@umich.edu (V.L.); wdhawk@umich.edu (W.D.H.); klionsky@umich.edu (D.J.K.); 2Department of Molecular, Cellular and Developmental Biology, University of Michigan, Ann Arbor, MI 48109, USA; 3Department of Pathology, Michigan Medicine, University of Michigan, Ann Arbor, MI 48109, USA; da_veiga_leprevost.felipe@gene.com (F.d.V.L.); nesvi@med.umich.edu (A.I.N.)

**Keywords:** autophagy, crosslinking mass spectrometry, intrinsically disordered protein region, ubiquitin-like conjugation system

## Abstract

The Atg12 protein in yeast is an indispensable polypeptide in the highly conserved ubiquitin-like conjugation system operating in the macroautophagy/autophagy pathway. Atg12 is covalently conjugated to Atg5 through the action of Atg7 and Atg10; the Atg12–Atg5 conjugate binds Atg16 to form an E3 ligase that functions in a separate conjugation pathway involving Atg8. Atg12 is comprised of a ubiquitin-like (UBL) domain preceded at the N terminus by an intrinsically disordered protein region (IDPR), a domain that comprises a major portion of the protein but remains elusive in its conformation and function. Here, we show that the IDPR in unconjugated Atg12 is positioned in proximity to the UBL domain, a configuration that is important for the functional structure of the protein. A major deletion in the IDPR disrupts intactness of the UBL domain at the unconjugated C terminus, and a mutation in the predicted α0 helix in the IDPR prevents Atg12 from binding to Atg7 and Atg10, which ultimately affects the protein function in the ubiquitin-like conjugation cascade. These findings provide evidence that the IDPR is an indispensable part of the Atg12 protein from yeast.

## 1. Introduction

Macroautophagy, hereafter autophagy, is an evolutionarily conserved pathway that cells utilize to maintain homeostasis. This nutrient-recycling catabolic process targets damaged organelles and superfluous cytoplasmic material for degradation. A repertoire of proteins from the autophagy machinery produces a cup-shaped structure, the phagophore, that is expanded until it ultimately engulfs the cytoplasmic cargo in the transient, double-membrane autophagosome. Fusion of the autophagosome with the vacuole (in yeast or plants) or lysosome (in cells from more complex eukaryotes) ensures breakdown of the unnecessary material into reusable components, which are effluxed from the degradative organelle back to the cytosol [1].

Expansion of the phagophore membrane relies on conjugation of Atg8 (LC3 or GABARAP in more complex eukaryotes) to the primary amino head group of phosphatidylethanolamine (PE). This ubiquitination-like reaction is mediated by an E1-E2-E3-like enzymatic cascade, where Atg7/ATG7 (yeast/mammals), Atg3/ATG3, and the Atg12–Atg5-Atg16/ATG12–ATG5-ATG16L1 complex function as the E1, E2, and E3 ligase, respectively [2,3]. In yeast, the Atg12–Atg5-Atg16 complex localizes on the surface of the phagophore membrane [4], where Atg12 is covalently linked to Atg5. Prior to formation of this covalent bond, Atg12 transitions through the E1-E2-like enzymatic cascade. Specifically, the C-terminal Gly186 in Atg12 forms a temporary thioester bond with Cys507 in the activation domain (AD) of Atg7, the E1-like activating enzyme utilizing ATP to structurally activate Atg12. Gly186 of Atg12 is then transferred to Atg10 where it forms the thioester bond with Cys133 of Atg10. Finally, Atg10 as the E2-like conjugation enzyme mediates the isopeptide linkage of G186 in Atg12 to Lys149 in Atg5. The Atg12-free surface of Atg5 in the Atg12–Atg5 conjugate binds noncovalently to Atg16 [3], which is a protein containing a coiled-coil domain that mediates functionally important homodimerization [5,6,7]. The Atg12–Atg5-Atg16 complex interacts with Atg17 to target the E3-like enzyme to the phagophore assembly site (PAS), which promotes autophagosome formation and Atg8–PE conjugation [8]. The ubiquitin-like enzymatic cascade is highly conserved from yeast to humans, and deletion of a gene encoding a protein involved in this cascade is detrimental for autophagy in all eukaryotes [9,10].

Major components of the yeast ubiquitin-like conjugation cascade, such as Atg3, Atg4, Atg5, Atg7, Atg8, Atg10, and Atg16, have had their domains biochemically characterized and structurally visualized [11,12,13,14,15,16,17,18,19,20], except for Atg12. The structure of the Atg12 protein has been elusive over a large portion of its amino acid sequence. Atg12/ATG12 carries a C-terminal ubiquitin-like domain (UBL) that forms a main interface in the interaction with Atg7/ATG7, and is involved in conjugation to Atg5/ATG5, and binding to a long flexible loop of Atg3/ATG3 [15,16]. However, the N-terminal domain preceding the UBL domain remains largely uncharacterized. In fact, an earlier study concluded that the N terminus of Atg12 from yeast is not needed for protein function in autophagy [21], although a later study suggested its role in the recruitment of the Atg12–Atg5-Atg16 complex to the PAS [8]. Two crystal structures that are currently available in the protein database visualize only the UBL domains of yeast Atg12 (PDB ID: 3W1S) or human ATG12 (PDB ID: 4NAW); the N-terminal domains are omitted, due to non-coherent X-ray scattering. Thus, information on the N-terminal domain of Atg12/ATG12 is very limited, although it represents ~54%/36% of the protein, respectively. 

AlphaFold [22], a deep-learning based predictor of protein structure, has predicted the N-terminal domain of Atg12 as a low-confidence structure-free region. Low-confidence regions annotated by AlphaFold overlap with amino acid sequences that are predicted to be intrinsically disordered protein regions (IDPRs) [23], which are protein domains lacking a well-defined 3D structure [24,25,26,27]. IDPRs are essential in many signaling pathways [28,29,30,31], including autophagy [32,33,34,35]. Furthermore, numerous human diseases have an origin in dysfunctional IDPRs, making them an important therapeutic target [36,37,38,39,40,41]. The dynamic nature of IDPRs allows for a prompt protein conformational change that accommodates switching of binding partners [28,42]. IDPRs, lacking secondary and tertiary structure, are sensitive to proteolytic enzymes. Nascent Atg12 from yeast, unconjugated to any member of the ubiquitin-like conjugation cascade, has the C-terminal Gly186 accessible at a disordered tail. Here, we show that the Atg12 protein needs its N-terminal IDPR not only for protection of the C-terminal tail against proteolysis but also for a functional conformation that efficiently interacts with Atg7 or Atg10. Based on consideration of all the data shown in this study, we present a presumptive model of the unconjugated Atg12 molecule, and we show how this molecule can interact with Atg7 and how it can noncovalently bind to Atg10.

## 2. Results

### 2.1. Atg12 Architecture and Evolutionarily Conserved Elements in the N Terminus of Atg12

Analysis of the amino acid sequence of Atg12 from yeast by multiple predictors for protein intrinsic disorder shows a hybrid architecture, in consensus with AlphaFold. The N-terminal domain of Atg12 is an intrinsically disordered protein region (IDPR) spanning amino acid residues 1-100. The UBL domain (amino acid residues 101-183) is well-folded, as expected (Figure 1A). The UBL domain is not the only conserved region in the Atg12 amino acid sequence. The IDPR carries two homologous segments (Appendix A) that overlap with putative binding regions predicted by the ANCHOR algorithm (Figure 1A). Because disordered protein sequences cannot fold and be stabilized on their own, this algorithm is designed to score for the intrinsically disordered segments that can be stabilized only via interaction with a globular binding partner. It is the involvement of ANCHOR regions in protein interactions that render their functional importance and drive evolutionary homology (Figure 1A and Appendix A).

### 2.2. The IDPR Is Important for Protein Function in Nonselective Autophagy 

To assess the functional importance of the evolutionarily conserved elements in the IDPR of Atg12, we introduced mutations in homologous amino acid residues within the two ANCHOR regions, specifically, the L5E E8K double mutation in ANCHOR1 and L54E L57E in ANCHOR2 (Figure 1A and Appendix A), to disrupt hydrophobic and/or electrostatic interactions mediated by these elements. Mutagenesis of conserved serine residues was omitted due to the lack of information on regulation of Atg12 via phosphorylation. We also created a deletion mutant (∆3-79) that removed a major portion of the IDPR including both ANCHOR regions. The F185E mutation at the C terminus was used as a negative control, as Glu in place of Phe presumably interferes with the insertion of the C-terminal tail into a hydrophobic cavity in the activation domain of Atg7, analogous to the characterized Atg7–Atg8 interaction ([17]; PDB ID: 3VH3). The function of the above-described mutants in nonselective autophagy was probed for by the prApe1 processing assay, where the transport of the precursor form of the vacuole-resident hydrolase Ape1 to the vacuole under starvation conditions is monitored in cells whereby the Cvt pathway is blocked by *vac8* deletion. A defect in prApe1 maturation is manifested by a decreased prApe1:total Ape1 ratio. Detection of prApe1 maturation in *atg12*∆ *vac8*∆ cells expressing Atg12 variants showed that the L5E E8K mutant behaved similar to the wild type, whereas the L54E L57E and deletion mutants were markedly defective in nonselective autophagy and F185E exhibited, as expected, a complete block in prApe1 processing (Figure 1B). This result suggests that the ANCHOR2 element is functionally more relevant than ANCHOR1.

### 2.3. The Atg12 IDPR Does Not Associate the Protein with Cellular Membranes

Finding that Atg12 IDPR is indispensable for efficient nonselective autophagy (Figure 1B) prompted us to further investigate the role of this region. Many proteins of the autophagy machinery have a domain that allows them to interact with cellular membranes [43,44]. We asked whether the Atg12 IDPR is such a domain. To probe a role of the Atg12 N terminus in membrane association, we carried out subcellular fractionation experiments using multiple-knockout (MKO) cells, which lack most of the *ATG* genes including those involved in the Atg8 and Atg12 conjugation systems (Appendix A), and which express the His_6_-Atg12 fusion protein. We found that approximately half of the wild-type population was associated with the membrane fraction whereas the other half remained cytosolic (Appendix A). A similar distribution of populations was observed for the deletion and L54E L57E mutants. In contrast, the IDPR alone (residues 1-100) was exclusively cytosolic (Appendix A), which indicates that this region does not have a membrane-associating capability. 

To further confirm this result, we constructed a protein chimera expressing the N-terminal region of Atg12 at the N terminus of Atg8. Atg8 is another UBL domain-containing substrate for the E1-E2-E3-like enzymatic cascade operating in autophagy. This cascade is missing in MKO cells, which prevents the conjugation of Atg8 to PE on the membrane. Thus, Atg8 in MKO cells can be used for probing the capability of this ubiquitin-like molecule to associate with membranes. Subcellular fractionation experiments with the Atg8 wild-type and chimeric protein showed that the wild-type population of Atg8 expressed in MKO cells was, as in the case of Atg12, split between the membrane and cytosolic fraction, whereas the Atg12[1-100]Atg8 protein was exclusively cytosolic (Appendix A). This result shows that the Atg12[1-100] peptide prevented membrane localization of Atg8. Together, the data in Appendix A exclude any role of the N terminus in membrane localization of Atg12. It is currently unknown whether membrane localization of Atg8 and Atg12 in MKO cells is direct or mediated by an undiscovered protein that was not knocked out in the MKO cells. Future research will hopefully explain this observation that is beyond the scope of the present study. 

### 2.4. The IDPR Is Positioned near the UBL Domain of Atg12

Atg12 migrates on SDS-PAGE gels as a compact protein without proteolysis at the calculated molecular mass of 22 kDa (Figure 1B). This finding suggests that the Atg12 IDPR is positioned close to the UBL domain, otherwise the IDPR would be expected to be sensitive to cellular proteases. To probe this hypothesis, we co-expressed Atg12 with Atg7 and Atg10 in *E. coli* and applied crosslinking mass spectrometry (XL-MS) on partially purified recombinant His_6_-Atg12 (Appendix A) crosslinked with BS^3^, a crosslinker that bridges primary amines of lysine residues with a spacer length of 11.4 Å. Two replicates of the XL-MS analysis (Appendix A) yielded a reproducible set of crosslinked lysines from the IDPR and UBL domains, suggesting that these domains interact or at least are in close proximity to each other. Specifically, in one experiment, Lys91, Lys93, Lys96, and Lys99 in the IDPR crosslinked with Lys103 and Lys118 in the UBL domain (Figure 2A,B). In the other experiment, Lys91, Lys93, Lys96, and Lys99 crosslinked with Lys103 (Appendix A). XL-MS also yields data on monolinks, which are one-sided attachments of a crosslinker. Monolinks provide information on amino acid residues that are accessible, not buried within a protein structure. In our XL-MS, BS^3^ monolinks were found on Lys84, Lys91, Lys93, Lys96, Lys99, Lys103, Lys107, Lys118, and Lys138 (Figure 2C). Because Lys84, Lys107, and Lys 138 were found only in the BS^3^ monolinks, these three amino acid residues are surface-accessible but likely at a distance longer than those crosslinkable by BS^3^.

### 2.5. Intramolecular Interactions between the IDPR and UBL Domain Secure the Intact Structure of Unconjugated Atg12

Finding that the IDPR is in close proximity to the UBL domain prompted us to probe intramolecular interactions in more detail. For this purpose, we used deletion mutants lacking one or both ANCHOR elements. The His_6_-Atg12[∆3-79] mutant, lacking both ANCHOR segments, exhibited multiple cleavage isoforms on western blots; this result occurred in MKO yeast cells as well as in *atg12*∆ *vac8*∆ cells, but was more easily detectable in the latter (Figure 1B and Appendix A). In contrast, wild-type His_6_-Atg12 appeared to be intact because it was detected by western blotting as a single band at 22 kDa (Figure 1 and Appendix A). The calculated molecular mass of His_6_-Atg12[∆3-79] is 13.5 kDa, but the major protein form detected by anti-His antibody exhibited a size of no more than 12 kDa, suggesting that a peptide of approximately 1.5 kDa (the N_172_-G_186_ segment) may be missing at the C terminus of His_6_-Atg12[∆3-79]. Mapping this cleaved-off segment onto the AlphaFold model of Atg12 revealed that the Atg12 UBL domain may be shorter in the deletion mutant. To explore this phenomenon, we constructed a double-tagged His_6_-Atg12-FLAG protein, and tested detection of the wild type and the ∆3-79 variant by western blotting using anti-His and anti-FLAG antibodies. We hypothesized that, in contrast to His_6_-tag detection, the anti-FLAG antibody would detect only WT but not the deletion mutant if the entire population of His_6_-Atg12[∆3-79]-FLAG had the C terminus cleaved off. However, if there was a small subpopulation of His_6_-Atg12[∆3-79]-FLAG with an intact C terminus, the anti-FLAG antibody would detect this species but only as a single band on the western blot; the multiple-band pattern would disappear. The anti-FLAG signal in our experiments (Figure 3A–D) revealed a detectable single band, demonstrating that there was a small subpopulation of His_6_-Atg12[∆3-79]-FLAG in *atg12*∆ *vac8*∆ cells that escaped proteolytic cleavage and retained the intact C terminus (Figure 3E). Taking into account the structural plasticity of IDPRs, a conformational ensemble of the truncated Atg12 IDPR swinging near the C terminus in this subpopulation can be seen as a cloud occluding the access of proteases. Conversely, detection of the multiple-band pattern for His_6_-Atg12[∆3-79] and His_6_-Atg12[∆3-79]-FLAG with anti-His antibody shows that a significant population of ∆3-79 had the C terminus cleaved off (Figure 3A–C). Affinity isolation using the His tag confirmed that ∆3-79 had isoforms of various lengths (Figure 3D). This result shows that the IDPR of Atg12 shields the C-terminal tail in the UBL domain and provides protection against proteolysis of the unconjugated C terminus. Regulation of such a temporary shielding, until the C-terminal tail in Atg12 enters the hydrophobic cavity of Atg7, remains elusive. Nevertheless, our data indicate that plasticity of the N-terminal IDPR is an important physiological determinant of the Atg12 structure.

Because dynamic disordered protein regions cannot fold into a secondary or tertiary structure on their own, the IDPR of monomeric full-length Atg12 can only protect the C terminus by being held in a shielding position via intramolecular interactions with the UBL domain surface. The ANCHOR elements are two segments where the IDPR can attach on the surface of the UBL domain, as they are, by definition, the segments stabilizing disordered regions via binding to a globular protein. In our next experiments we focused on a part of the Atg12 N terminus carrying ANCHOR1. Given the dispensability of ANCHOR1 for protein function in nonselective autophagy (Figure 1B), we asked whether ANCHOR1 has a structural contribution in the N terminus of unconjugated Atg12. Specifically, we wanted to determine whether ANCHOR1 cooperates with ANCHOR2 and other intramolecular contacts to secure the position of the IDPR in proximity to the UBL domain, and thereby preventing proteolysis of the disordered N terminus of Atg12. To make this determination, the ANCHOR1 region was removed by deleting the amino acid residues 3-27 in the single-tagged and double-tagged Atg12. We reasoned that if the IDPR in the deletion mutant became detached from the UBL domain and susceptible to proteolysis at the N terminus then His_6_-Atg12[∆3-27] and His_6_-Atg12[∆3-27]-FLAG would not be detectable by western blotting with anti-His antibody (Figure 3F). In contrast, the anti-FLAG antibody would detect His_6_-Atg12[∆3-27]-FLAG if the deletion of ANCHOR1 did not have a far-reaching effect on the C terminus. However, if the IDPR remained protected from proteolysis by proximity to the UBL domain surface in the absence of ANCHOR1 (Figure 3F), then His_6_-Atg12[∆3-27] and His_6_-Atg12[∆3-27]-FLAG would be detected by western blotting with both anti-His and anti-FLAG antibodies. Our data obtained on whole-cell lysates from nitrogen-starved *atg12*∆ *vac8*∆ cells expressing His- and FLAG-tagged variants of Atg12 showed a clear detection of proteins with both antibodies (Figure 3G,H). This result shows that ANCHOR1 is not structurally essential for wrapping of the IDPR on the unconjugated UBL domain and that there are other intramolecular interactions that secure the IDPR-UBL domain configuration. A slightly weaker signal for the Atg12[∆3-27] variant relative to the full-length version indicates that ANCHOR1 may marginally contribute to the overall stability of the unconjugated protein (Figure 3F–H). Future research will hopefully reveal whether there is a particular role of ANCHOR1, perhaps in an autophagy-independent function, which was found for human ATG12 [45].

### 2.6. The Putative α0 Helix in the IDPR Is Needed for Binding of Atg12 to Atg7 

Atg7 is the first protein in the ubiquitin-like conjugation cascade that interacts with nascent Atg12. We used affinity isolation to explore the involvement of the Atg12 N terminus in this intermolecular protein–protein interaction. The *atg7*∆ *atg12*∆ cells were transformed with plasmids that allowed for overexpression of PA-Atg7 and His_6_-Atg12 under the control of the *CUP1* promoter. Affinity-isolation experiments showed that full-length PA-Atg7 interacted efficiently with His_6_-Atg12 (Figure 4A). The Atg12 UBL domain adopts a fold that is very similar to that of Atg8 (Figure 4B), which is why both proteins were proposed to interact with Atg7 in a very similar fashion [12]. A previous study focusing on the Atg7–Atg8 interaction showed a two-step binding process. In the first step, the disordered C-terminal tail (amino acids 601-630) of Atg7 is necessary for efficient initial recognition of the Atg8 substrate ([17]; PDB ID 2LI5). In the second binding step, a portion of the Atg7 tail folds into the α17-helix (residues 601-613) and together with the activation (AD; residues 295-572) and extreme C-terminal (ECTD; residues 573-600) domains of Atg7 creates an interface between Atg7 and the Atg8 UBL domain ([17]; PDB ID 3VH3) (Figure 4B). We asked if the same domains of Atg7 are also required for binding to Atg12. For this purpose, we constructed two Atg7 mutants, one carrying a deletion of the disordered C-terminal tail spanning amino acids residues 601-630 and the other lacking the folded AD and ECTD (residues 295-600). Affinity-isolation experiments (Figure 4A) showed that the deletion variants of Atg7 did not interact with Atg12 as efficiently as full-length Atg7, suggesting that both domains of Atg7 play a role in the Atg7–Atg12 interaction.

The initial binding of the Atg7 disordered C-terminal tail to Atg8 involves the Atg8 α1 and α2 helices in a short N-terminal helical domain upstream of the Atg8 UBL domain ([17]; PDB ID: 2LI5). The AlphaFold model of Atg12 showed a single α helix within the IDPR (Figure 2B), termed here as α0. The putative α0 helix overlaps with the ANCHOR2 element, which indicates that α0 may be stabilized via attaching to the UBL domain [46]. We asked whether Atg12 α0 is involved in the Atg7–Atg12 interaction, as Atg8 α1 and α2 are in the Atg7–Atg8 interaction. Helices are typically disrupted by replacing a hydrophobic amino acid residue with a negatively charged residue. We used the initially characterized L54E L57E mutant (Figure 1B and Appendix A) and compared it to the wild type. Affinity isolation showed that the L54E L57E mutant failed to co-immunoprecipitate with Atg7 (Figure 4C). Introduction of the C507S mutation into Atg7, which replaces the thioester bond between the Atg7 catalytic Cys507 and Atg12 C-terminal Gly186 with a more stable ester bond, did not improve co-immunoprecipitation of L54E L57E relative to the wild type (Figure 4D), indicating that the mutation-induced defect in Atg12 could not be eliminated by a stronger covalent bond between Atg7 and Atg12, likely occurring before the bond can be formed. Together the results in Figure 4 show that the ANCHOR2 segment overlapping with the putative α0 of Atg12 is required for the efficient interaction between Atg7 and Atg12, where Atg12 α0 appears to play a role similar to that of Atg8 α1 and α2.

### 2.7. Atg12 Interacts with Unconjugated Partial Atg10 in an IDPR-Dependent Manner

In the ubiquitin-like conjugation cascade, Atg12 is transferred from Atg7 (E1-like enzyme) to Atg10 (E2-like enzyme). Specifically, the C-terminal Gly186 of Atg12 forms a thioester bond with Cys133 of Atg10. It is plausible that this bond is not the only contact between Atg10 and Atg12 and that both protein surfaces establish additional noncovalent contacts in their interface. A crystal or NMR structure of the covalent Atg10–Atg12 complex is not available to see these putative additional contacts. Probing these contacts in the covalent complex by other methods requires consideration of a few aspects. First, the presence of the thioester Atg10–Atg12 bond can obscure detection of a mutation-induced disruption in noncovalent contacts. Second, if residues involved in the noncovalent Atg10–Atg12 interface are also involved in the Atg7–Atg12 interaction then the noncovalent Atg10–Atg12 contacts cannot be probed in the frame of the covalent Atg10–Atg12 complex, as this complex is not formed (i.e., the first defect has an epistatic effect). The second aspect is the case of Atg12^L54,57E^ that binds poorly to Atg7 (Figure 4). To find a role of the N-terminal Atg12 IDPR in the Atg10-Atg12 noncovalent interface, we tested binding between unconjugated Atg10 and Atg12. We applied affinity isolation on *atg7*∆ *atg10*∆ *atg12*∆ cells overexpressing PA-Atg10 and His_6_-Atg12 under the control of the *CUP1* promoter. Our data show (Figure 5A) that PA-Atg10 co-immunoprecipitated His_6_-Atg12, suggesting that Atg10 stably interacts with Atg12 in the absence of their covalent bond. Probing of the Atg12^L54,57E^ mutant in this interaction revealed that the L54E L57E mutation in the putative α0 helix abolished binding between unconjugated Atg10 and Atg12 (Figure 5A). This result shows that the Atg12 IDPR is necessary in the noncovalent Atg10-Atg12 interface. 

The affinity-isolation experiment (Figure 5A) also revealed an intriguing observation with regard to the noncovalent Atg10-Atg12 complex. In the absence of conjugation to Atg12, overexpressed Atg10 was susceptible to proteolysis at the C terminus. In particular, PA-Atg10 with the 14-kDa PA tag migrated on an SDS-PAGE gel at about 27 kDa, instead of the expected 34-kDa polypeptide, suggesting that His_6_-Atg12 overexpressed in yeast cells binds partial Atg10 of about 13 kDa, instead of the 20-kDa full-length protein. This finding indicates that, unless the protein is conjugated to Atg12, Atg10 lacks a C-terminal region of about 7 kDa, theoretically pointing to proteolysis near F112. Mapping this region on the crystal structure of Atg10 from yeast (light orange; [11]; PDB ID: 4EBR) reveals the putative part of the Atg10 protein (dark orange) that interacts with full-length Atg12 (Figure 5B). 

To test this hypothesis, we constructed a truncation mutant of Atg10 that removes amino acid residues 111-167. Western blot analysis shows that Atg10[∆111-167] is poorly detectable compared to wild type (Figure 5C), suggesting that the mutant may not fold properly without the C-terminal region. Nevertheless, Atg10[∆111-167] migrated at about 26 kDa, only 1 kDa less than the presumably cleaved wild type (Figure 5C). This gel migration of both Atg10 variants close to each other demonstrates that the wild type in the Atg12-unconjugated form is likely to correspond to partial Atg10 lacking its C-terminal region. Figure 5C shows that Atg10 proteolysis at the C terminus occurs independently of the absence or presence of Atg7 or Atg12 in the yeast cell. His_6_-Atg12 co-immunoprecipitated partial PA-Atg10 along with PA-Atg7 (Figure 5D), further confirming that Atg7 cannot prevent C-terminal proteolysis of conjugation-free Atg10. Unless Atg10[∆111-167] undergoes additional proteolysis at the C terminus, proteolytic cleavage of wild-type Atg10 occurs most likely in the loop a few residues downstream of F112 (Figure 5B). In contrast, Atg10 in the Atg12-conjugated form maintained its full length, as evidenced by the detection of the Atg10–Atg12 conjugate at about 56 kDa (Atg10, 20 kDa; Atg12, 22 kDa; PA tag, 14 kDa) (Figure 5D). Testing whether the Atg10[∆111-167] mutant was able to interact with Atg12 by affinity isolation showed that PA-Atg10[∆111-167] co-immunoprecipitated His_6_-Atg12 in an IDPR-dependent manner, although a good evaluation of this interaction was significantly obscured by very different cellular levels of PA-Atg10[∆111-167] and free PA (Appendix A).

In support of the results obtained from yeast cells, we verified using recombinant proteins from *E. coli* that Atg12 noncovalently bound partial Atg10 (Appendix A). His_6_-Atg12 was co-expressed with Atg7 and Atg10 in bacterial cells and pulled down using Ni-NTA agarose. The imidazole eluate was applied onto an anion exchange column. The fraction most enriched in recombinant His_6_-Atg12 was then separated by size-exclusion chromatography (SEC; Appendix A). Analysis of the His_6_-Atg12-containing fractions after anion exchange and SEC showed that recombinant Atg12 co-purified with a polypeptide of ~13 kDa that was detected by western blot with anti-Atg10 antibody (Appendix A), which should detect a band at about 20 kDa if Atg10 was a full-length protein. Mass spectrometry analysis (Appendix A) of the protein band at ~13 kDa (green rectangle) positively identified Atg10 by five unique peptides and 50% sequence coverage. It needs to be noted that the SEC fractions B12 and B11 exhibited a total molecular mass of 55 kDa, meaning that Atg7 of 70 kDa cannot be in these fractions and thus partial Atg10 co-purified from bacterial lysates with Atg12 independent of Atg7, as observed with in vivo yeast samples. Finally, detection of the Atg10–Atg12 conjugate at 42 kDa by anti-Atg12, anti-His, and anti-Atg10 antibodies (Appendix A) demonstrated that only covalent conjugation to Atg12 could protect the full length of Atg10, as seen in yeast (Figure 5D).

### 2.8. Disruption of the IDPR by the L54E L57E Mutation Decreases Efficiency of the Cvt Pathway and Atg8 Lipidation

We assumed that the disruption of the IDPR of Atg12 that interferes with the Atg7–Atg12 and Atg10–Atg12 interactions may have an impact on Atg12 function. To probe this assumption, we analyzed the Cvt pathway under nutrient-rich conditions and Atg8 lipidation after 4 h of nitrogen starvation. The Atg12 wild type and L54E L57E mutant were expressed under the control of the native or *CUP1* promoter in *atg12*∆ cells. Both assays revealed that L54E L57E exhibited significant defects in the Cvt pathway and Atg8–PE conjugation under growing and starvation conditions, respectively (Figure 6A,B), suggesting that the intact ANCHOR2-α0 in Atg12 is important for the efficient function of the protein.

## 3. Discussion

A major portion of Atg12 from yeast is comprised of the IDPR upstream of the UBL domain. Yet, function–structure relationships for this region are elusive and its role in the interactions of Atg12 with Atg7 or Atg10 has not been explored. In this study, we revealed that the IDPR of Atg12 is important for the functional structure of the unconjugated protein. The XL-MS data combined with biochemical analyses show that the IDPR is positioned near the UBL domain, in contrast to visualization by the AlphaFold model, where the IDPR is freely detached and appears to be floating far from the UBL domain (Figure 2B,C). A stable configuration of the IDPR in proximity to the UBL domain is mediated by intramolecular interactions involving the highly conserved ANCHOR regions (Figure 1A and Appendix A) [46]. The conformational ensemble of the IDPR near the UBL domain ensures protection of the disordered N and C termini against proteolysis and grants the protein a compact conformation that migrates on SDS-PAGE gel without aberration (Figure 7A, left). However, this migration becomes aberrant when the mutation L54E L57E is introduced into the putative α0 helix within the ANCHOR2 region (Figure 1, Figure 2, Figure 4, Figure 5, Figure 6 and Appendix A). In particular, the mutant migrates more slowly than the wild type, which runs at the calculated molecular mass of 22 kDa. This slow gel migration phenomenon is a typical hallmark of glutamic/aspartic residues exposed on a protein surface, which results in poor SDS binding [47]. Thus, L54E L57E migration can be interpreted to indicate that E54 and E57 are exposed on the Atg12 surface (Figure 7A, right), whereas L54 and L57 in the wild type tend to be in a hydrophobic environment. Because disordered regions cannot fold on their own into a secondary or tertiary structure, L54 and L57 can only be in a hydrophobic interface induced by the IDPR attached on the UBL domain surface. It is interesting to note that the C terminus in L54E L57E remained protected from proteolysis, as no multiple isoforms were detected by western blotting, which indicates that the mutation caused only a local unfolding around the ANCHOR2 region (Figure 7A, right).

Nascent Atg12 enters the ubiquitin-like conjugation cascade via binding to Atg7. The Atg7–Atg12 interaction has not been structurally or mechanistically studied, in contrast to the Atg7–Atg8 interaction [12,17,48]. The latter has two phases. In the first phase, the C-terminal disordered tail of Atg7 binds the Atg8 surface composed of α1, α2, β2, and α3 [17]. This initial Atg7–Atg8 complex utilizes three binding pockets on Atg8, the W and L site, also recognized by the Atg8-family interacting motif/AIM, and the third pocket created between the α1 and α2 helices in the Atg8 N-terminal helical domain. The N-terminal helical domain upstream of the Atg8 UBL domain is disordered in unprocessed Atg8 and becomes helical after processing by Atg4 [49]. Thus, the IDPR of Atg8 is important for the formation of the initial Atg7–Atg8 complex. In the second binding phase, a portion of the C-terminal tail of Atg7 folds into α17 and thereby brings the Atg8 molecule close to the AD of Atg7. This allows the Atg8 UBL domain to create multiple contacts with the Atg7 AD and slide its C-terminal Gly116 into the hydrophobic cavity for activation [17,48]. The biochemical data presented here suggest that Atg7 may follow similar steps when it interacts with Atg12 (Figure 7B). We showed that Atg7 needs the C-terminal tail as well as the AD for efficient binding to Atg12. Furthermore, similar to the IDPR of Atg8, the IDPR of Atg12 is indispensable for Atg7–Atg12 binding, as disruption by the L54E L57E mutation in the putative α0 helix abolishes this interaction. Our data are in agreement with a previous structural study [12], which proposed (but did not directly show) that the UBL domains of Atg8 and Atg12 (Appendix A) fit in a similar way into the Atg3-Atg7 and Atg10-Atg7 dimer, respectively.

When Atg12 G186 enters the hydrophobic cavity of the Atg7 AD, shielding of the Atg12 C terminus by the Atg12 IDPR is presumably released but the α0 helix in the IDPR of Atg12 is retained for stable binding to Atg10. The manner by which Atg10 interacts with Atg12 in the Atg10–Atg12 conjugated complex has not been explored. It is conceivable that both proteins create a noncovalent binding interface in support of their thioester bond. Our data (Figure 5A–D) are consistent with this assumption and show with unconjugated proteins that an Atg10 region upstream of F112 is part of the Atg10–Atg12 noncovalent binding interface, where the intact Atg12 IDPR is indispensable (Figure 7C).

## 4. Materials and Methods

### 4.1. Yeast Plasmids, Strains and Media

pAtg12-3HA-Atg12(pRS416), pCuHis_6_-Atg12(pRS424) under the control of the *CUP1* promoter, pCuHis_6_-Atg12-FLAG(pRS424), pCuHis_6_-Atg12[1-100](pRS424), pCuAtg12[1-90]Atg8(pRS426), and pCuPA-Atg7_CuPA-Atg10(pRS416) were generated using the FastCloning technique [50]. The pCuPA-Atg7 and pCuPA-Atg10 plasmids were kind gifts from Dr. Brenda Schulman (Max Planck Institute of Biochemistry). Other plasmids were generated by site-directed mutagenesis [51]. Yeast cells were grown in synthetic minimal medium (SMD; 0.67% yeast nitrogen base, 2% (w:vol) glucose, and auxotrophic amino acids and vitamins). Autophagy was induced by shifting the cells to nitrogen-starvation medium (SD-N; 0.17% nitrogen base without ammonium sulfate or amino acids, and 2% [w:vol] glucose). Strains used in this study are listed in Appendix A.

### 4.2. Yeast In Vivo Assays

The precursor aminopeptidase I (prApe1) processing and Atg8-lipidation assays were performed as described previously [52]. Briefly, cells grown in selective nutrient-rich medium were harvested (1 OD_600_ units) to probe the cytoplasm-to-vacuole targeting (Cvt) pathway. Nonselective autophagy and Atg8 lipidation were induced by shifting cells to nitrogen-starvation medium for 4 h. Cell pellets were washed with water and analyzed by western blotting using Ape1 (1532) or Atg8 (69-14) antisera. Densitometry quantification of the western blots was performed using ImageJ software version 1.52a.

### 4.3. Subcellular Fractionation

Cells (15 OD_600_ units) were lysed in 0.250 mL of PS200 buffer (20 mM PIPES-KOH, pH 6.8, 200 mM sorbitol, 5 mM MgCl_2_, 1 mM PMSF, cOmplete EDTA-free protease inhibitor) with glass beads in five cycles of vortexing for 45-s and 2-min intervals on ice. Cell debris were removed by centrifugation at 1000× *g* for 10 min, and 150 µL of supernatant was centrifuged at 13,000× *g* for 10 min to separate the cytosolic- and membrane-associated fraction. The resulting ~150 µL of supernatant was precipitated with a TCA–acetone mix (100 µL TCA, 800 µL acetone) as the cytosolic fraction. The precipitate was washed with acetone and air-dried. The air-dried sample and pellet containing the membrane-associated fraction were resuspended in 50 µL of SDS-PAGE loading buffer containing 4 mM PSMF and incubated for 15 min at 55 °C before analysis by western blotting. Pgk1 and Dpm1, which were used as cytosolic- and membrane-associated controls, respectively, were detected on the western blot using anti-Pgk1 antiserum (generously provided by Dr. Jeremy Thorner, University of California, Berkeley) and anti-Dpm1 monoclonal antibody (Invitrogen/Fisher Scientific, Pittsburgh, PA, USA, A6429). His_6_-Atg12 and Atg8 were detected by anti-polyhistidine monoclonal antibody (Sigma-Aldrich, St. Louis, MO, USA, H1029) and Atg8 (69-14) antisera, respectively. His_6_-Atg12-FLAG was detected with an anti-polyhistidine monoclonal antibody (Sigma-Aldrich, H1029) and anti-FLAG M2 monoclonal antibody (Sigma-Aldrich, F1804).

### 4.4. Affinity-Isolation Experiments

*S. cerevisiae* cells (50 OD_600_ units) were lysed in 0.5 mL of lysis buffer (PBS [137 mM NaCl, 2.7 mM KCl, 10 mM Na_2_HPO_4_, 2 mM K_2_HPO_4_, pH 7.4], 0.2 M sorbitol, 1 mM MgCl_2_, 0.1% or 0.5% Triton X-100, 1 mM PMSF, cOmplete EDTA-free protease inhibitor [Roche/Fisher Scientific, 11836170001]) with glass beads in six cycles of vortexing as described above. Cell debris were removed by centrifugation at 4000× *g* for 5 min. An aliquot of the supernatant fraction (20%) was TCA-precipitated as the input. The remaining supernatant was incubated with IgG Sepharose 6 Fast Flow (GE Healthcare/Fisher Scientific, 45-000-173) or TALON metal affinity resin (TaKaRa Bio USA, Ann Arbor, MI, 635501) for 2 h at 4 °C. After at least three washes with ice-cold lysis buffer, the proteins were eluted by incubation of the Sepharose at 55 °C for 15 min with SDS-PAGE buffer. For TALON resin elution, SDS-PAGE buffer was supplemented with 0.5 M imidazole. The eluted proteins were analyzed by western blotting with an anti-polyhistidine monoclonal antibody (Sigma-Aldrich, H1029) and anti-PAP antibody (Jackson Immunoresearch, West Grove, PA, USA, 323-005-024). 

### 4.5. Overexpression and Ni-NTA Purification of His_6_-Atg12 and BS^3^ Crosslinking

A DNA fragment encoding *S. cerevisiae* Atg12 (residues 1-186) was subcloned into pMSCG7 to generate His_6_-Atg12. *S. cerevisiae* Atg7 (residues 1-630) and Atg10 (residues 1-167) were subcloned into pCOLADuet-1 (a kind gift from Dr. S. Martens, University of Vienna). *Escherichia coli* BL21 Gold (DE3) cells (Agilent) transformed with both plasmids were grown in Terrific Broth medium at 37 °C to an OD_600_ of 0.6, induced with 1 mM isopropyl-beta-D-thiogalactopyranoside, and grown for 21 h at 19 °C. Cells were pelleted and stored at −80 °C. Pellets were thawed and resuspended in lysis buffer (PBS, pH 7.4, 1% Triton X-100, 1 mM PMSF, and cOmplete EDTA-free protease inhibitor). Cells were broken by sonication (Branson digital sonifier, Branson Ultrasonics Corp., Brookfield, CT, USA). Lysates cleared by centrifugation were mixed with Ni-NTA agarose (Qiagen/Fisher Scientific, 30210). The agarose was washed with washing buffer (PBS pH 7.4, 2.5 mM imidazole, 1 mM PMSF, and cOmplete EDTA-free protease inhibitor) and protein was eluted with elution buffer (PBS pH 7.4, 200 mM imidazole, 1 mM PMSF, and cOmplete EDTA-free protease inhibitor). The eluate was mixed with 0.4 M sucrose and stored at −80 °C until it was used for crosslinking analysis or further purification.

BS^3^ crosslinking took place on ice for 2 h in the protein eluate containing 5 mM BS^3^, and the reaction was stopped with 50 mM Tris-HCl, pH 7.5, at room temperature for 15 min. Proteins in samples were separated by SDS-PAGE, and the protein band corresponding to Atg12 was cut from the gel and submitted to crosslinking mass spectrometry. 

### 4.6. Anion Exchange and Size-Exclusion Chromatography

The Atg12-containing fraction eluted with 200 mM imidazole from Ni-NTA was applied to a HiTrap Q anion exchange column pre-equilibrated with 50 mM Tris, pH 8.0, and 100 mM NaCl. The column was washed with 50 mM Tris, pH 8.0, and 100 mM NaCl, and the protein was eluted using a salt gradient from 50–500 mM NaCl and then 500–1 M NaCl. Fractions containing the Atg12 protein were pooled and applied to a Superdex 75 size-exclusion chromatography column pre-equilibrated with PBS, pH 7.4, and 0.2 mM PMSF. Fractions containing Atg12 were pooled and analyzed. Proteins were detected on western blots by anti-polyhistidine antibody (Sigma-Aldrich, H1029), Atg12 antibody (Rockland/Fisher Scientific, 200-401-437), and Atg10 antisera.

### 4.7. Crosslinking Mass Spectrometry

The Atg12 protein in the gel was dissolved and digested with trypsin. The resulting peptide mixture was analyzed by liquid chromatography–tandem mass spectrometry (LC-MS/MS) as described in [53]. 

MS/MS spectra were analyzed using the pLink version 2.3.11 software [54] to identify crosslinked peptides. MS/MS spectra were searched against a FASTA protein sequence database containing yeast Atg12 and common contaminant proteins. The search parameters were set as follows: BS^3^ as the crosslinker, trypsin cleavage, allowing up to three missed cleavages, peptide mass between 500 and 6000, peptide length between 5 and 60, precursor and fragment tolerances set to 20 parts per million (ppm), fixed carbamidomethyl cysteine, variable methionine oxidation, 10 ppm filter tolerance, and 5% peptide spectrum match (PSM) false discovery rate (FDR). All identified crosslinks involving contaminant proteins were excluded. MS/MS spectra of crosslinked peptides were visualized and inspected using pLabel v2.4.1. 

### 4.8. Bioinformatics Analyses

The propensity for protein intrinsic disorder was predicted using the PONDR-FIT (URL: http://original.disprot.org/pondr-fit.php, accessed on 18 January 2018) IUPred2A (URL: https://iupred.elte.hu, accessed on 11 December 2019), and PrDOS (URL: https://prdos.hgc.jp/cgi-bin/top.cgi, accessed on 28 March 2020) predictors [55,56,57], and disordered binding regions were predicted using the ANCHOR algorithm (URL: http://anchor.elte.hu, accessed on 11 December 2019) [58]. Crystal structures were visualized using the PyMOL Molecular Graphic System, Version 2.0 Schrödinger, LLC. Amino acid sequences for multiple sequence alignments were obtained from the UniProt database. The alignment was created using the ClustalW Multiple alignment algorithm in the BioEdit program version 7.0.4.1. [59].

## 5. Conclusions

In this study, we present experimental evidence that the IDPR of Atg12 is not a dispensable protein domain with a random conformation arising from a low evolutionary homology. On the contrary, it is a region utilizing structural plasticity, binding promiscuity, and the ability to undergo a conformational switch near the UBL domain to fulfill its precise structural and functional purpose. The work shown here can be considered only the beginning on a long journey of discovering the multifaceted role of the Atg12 IDPR in the autophagy pathway. Knowledge on how the IDPR operates during the protein’s lifetime in yeast (Figure 7D) can be translatable and useful to studies of mammalian auto-phagy. The IDPR of human ATG12 is comprised of 51 amino acid residues. The NIH National Cancer Database has records of 43 somatic mutations in human ATG12 (https://portal.gdc.cancer.gov/genes/ENSG00000145782?ssmsTable_offset=20; accessed on 13 June 2023) and some of them are located within the ATG12 IDPR. Yet, conformation, configuration relative to the UBL domain, and function of this IDPR are unknown.

## Figures and Tables

**Figure 1 ijms-24-15036-f001:**
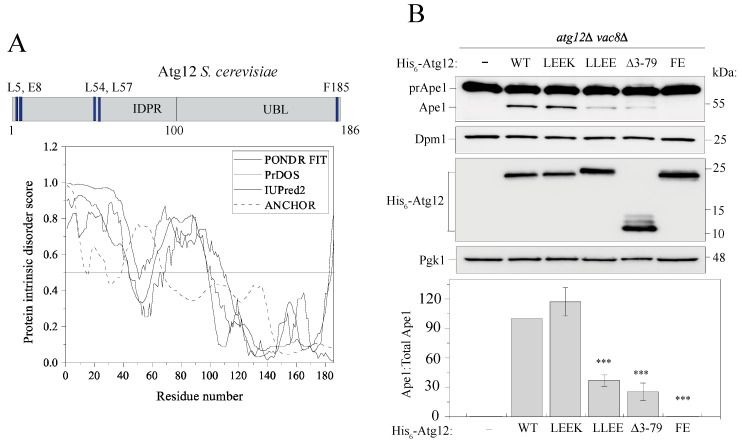
The N terminus of Atg12 is intrinsically disordered and is needed for nonselective auto-phagy. (**A**) Upper panel: schematic representation of Atg12 from *S. cerevisiae.* Lower panel; intrinsic order/disorder in Atg12 predicted by the PONDR-FIT (blue), PrDOS (green), and IUPred2 (black) algorithms, and the disordered binding domains predicted by the ANCHOR (dashed) algorithm. (**B**) Maturation of prApe1 measured as a readout of nonselective autophagy after the shift of *atg12*∆ *vac8*∆ cells to SD-N medium for 4 h. Cells were transformed with plasmids (pRS424, 2µ) carrying the indicated His_6_-Atg12 constructs; wild type (WT), L5E E8K (LEEK), L54E L57E (LLEE), and F185E (FE). Pgk1 and Dpm1 serve as loading controls. The Ape1:total Ape1 ratio was determined from three independent experiments. Error bars represent standard deviations. Statistical significance was tested using unpaired two-tailed Student’s *t* test: *** *p* < 0.0005.

**Figure 2 ijms-24-15036-f002:**
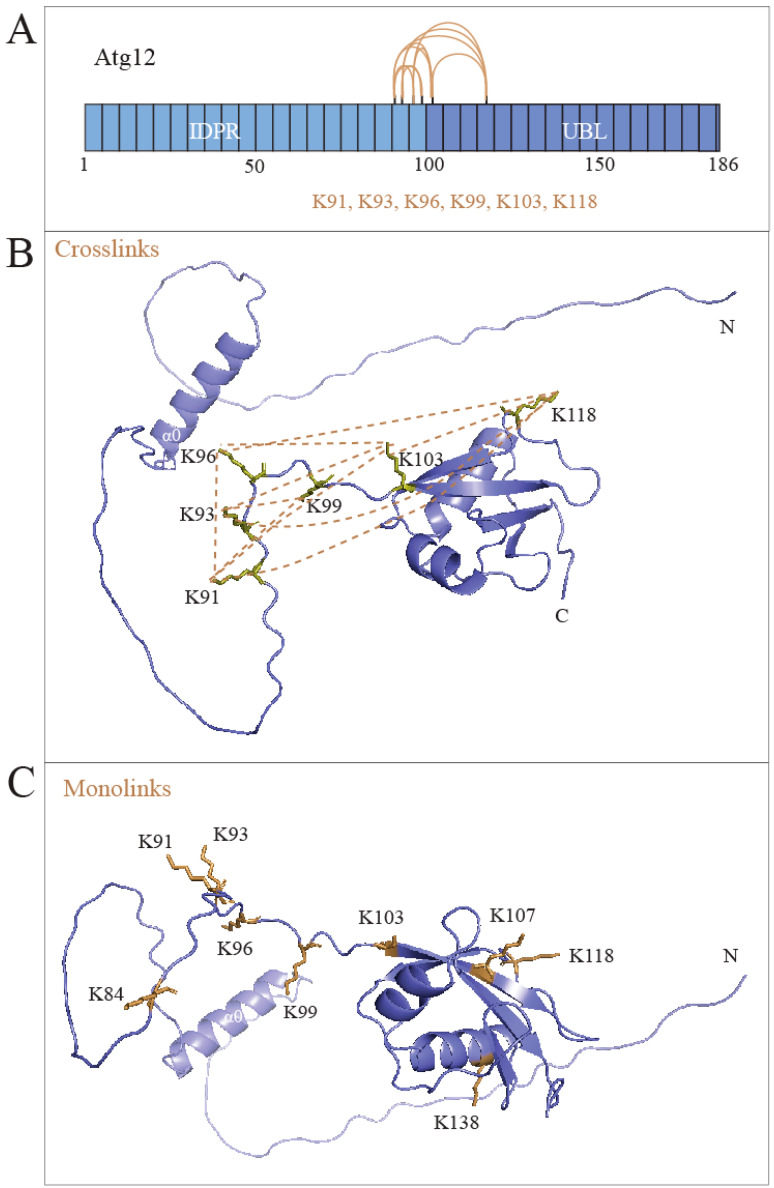
Intramolecular BS^3^ crosslinking in Atg12 revealed by XL-MS. (**A**) Visualization of intramolecular crosslinks by the BS^3^ crosslinker (brown) in the schematic representation of Atg12 (blue). Lysine residues involved in crosslinking are indicated below the scheme. (**B**) Mapping of BS^3^ crosslinks (dashed brown lines) on the AlphaFold model of Atg12 (blue). (**C**) Mapping of BS^3^ monolinks on the AlphaFold model of Atg12. Lysine residues (brown sticks) carrying the monolinked crosslinker are assumed to be surface-accessible.

**Figure 3 ijms-24-15036-f003:**
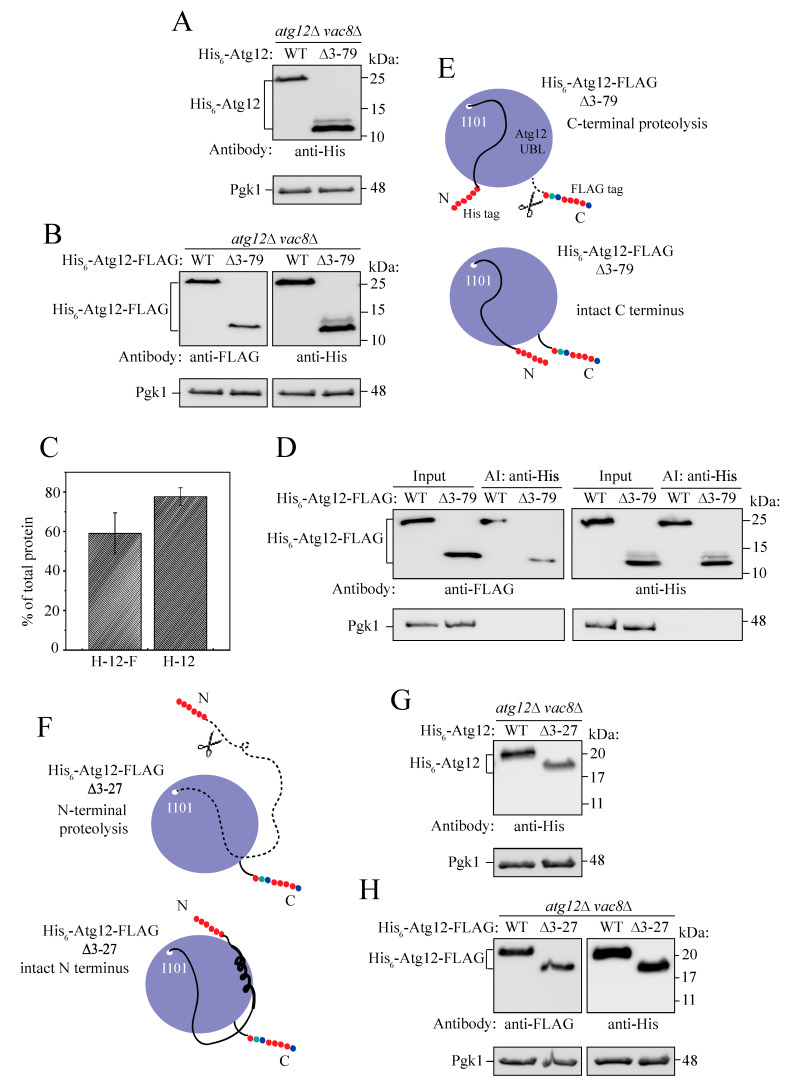
Intramolecular interactions in Atg12. (**A**–**C**) Western blot analysis of whole-cell lysates from *atg12*∆ *vac8*∆ cells collected after 4 h of nitrogen starvation. His_6_-Atg12, His_6_-Atg12[∆3-79] (H-12), His_6_-Atg12-FLAG and His_6_-Atg12[∆3-79]-FLAG (H-12-F) were overexpressed under the control of the *CUP1* promoter on the pRS424 (2µ) plasmid. The percentage of the major isoform in the total ∆3-79 protein detected by anti-His antibody was determined from three independent experiments. Error bars represent standard deviations. (**D**) Affinity isolation with *atg12*∆ *vac8*∆ yeast cells overexpressing His_6_-Atg12-FLAG wild type or ∆3-79 from the plasmids described in A-C. His_6_-Atg12-FLAG was pulled down by TALON metal affinity resin. Proteins were detected on western blots using anti-His, anti-FLAG, and anti-Pgk1 antibodies. (**E**,**F**) Schematic representations of the double-tagged Atg12 protein variants. N-terminal His_6_ tag, red dots; C-terminal FLAG tag, multicolor dots; UBL domain, blue sphere; IDPR, black line. I101 schematically marks the first amino acid residue in the UBL domain. (**G**,**H**) Western blot analysis of whole-cell lysates from *atg12*∆ *vac8*∆ cells collected after 4 h of nitrogen starvation. His_6_-Atg12[∆3-27] and His_6_-Atg12[∆3-27]-FLAG were overexpressed under the control of the *CUP1* promoter on the pRS424 plasmid.

**Figure 4 ijms-24-15036-f004:**
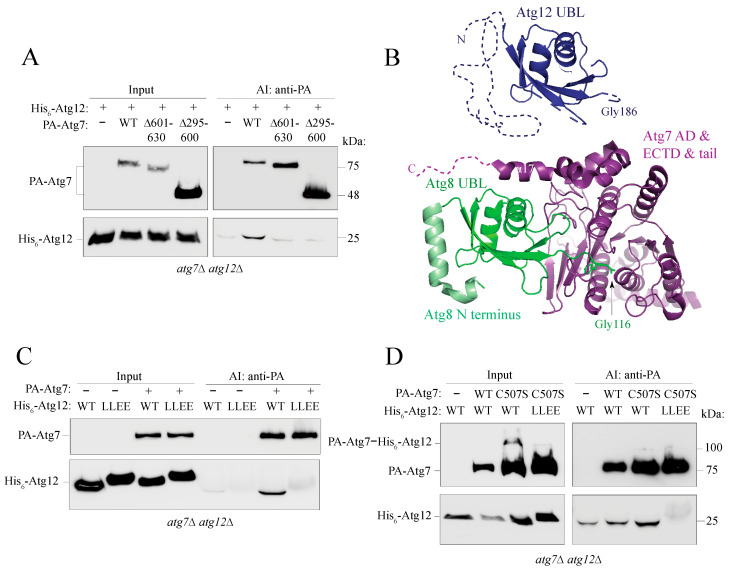
Probing the Atg7–Atg12 interaction. (**A**) Affinity isolation experiment with *atg7*∆ *atg12*∆ cells expressing PA-Atg7 and His_6_-Atg12 under the control of the *CUP1* promoter. Truncation variants of PA-Atg7 carry a deletion of the C-terminal disordered tail (∆601-630) or the folded adenylation (AD) and extreme C-terminal (ECTD) domains (∆295-600). (**B**) Crystal structure (IDB ID: 3VH3) of Atg8 (green) in the complex with Atg7 (AD + ECTD + tail; violet). The α17 helix (residues 601-613) folds from the C-terminal tail after pulling Atg8 to the AD. Atg8 is comprised of a short α-helical N terminus (light green) and the ubiquitin-like (UBL) domain (dark green). C-terminal Gly116 in Atg8 is activated and conjugated to Cys507 in the Atg7 AD. Atg12 (blue) is comprised of a long N-terminal IDPR (dashed line) and the UBL domain, visualized in the crystal structure (PDB ID: 3W1S; blue). Atg8 and Atg12 UBL domains share a very similar fold. (**C**) An affinity-isolation experiment with *atg7*∆ *atg12*∆ cells expressing full-length PA-Atg7 and either of the two His_6_-Atg12 variants (WT or L54E L57E [LLEE]) under the control of the *CUP1* promoter. (**D**) Affinity-isolation experiment as in C, except that PA-Atg7 is wild type or the C507S mutant. The Atg7 C507S mutation allows for formation of a more stable ester bond with Atg12 G186.

**Figure 5 ijms-24-15036-f005:**
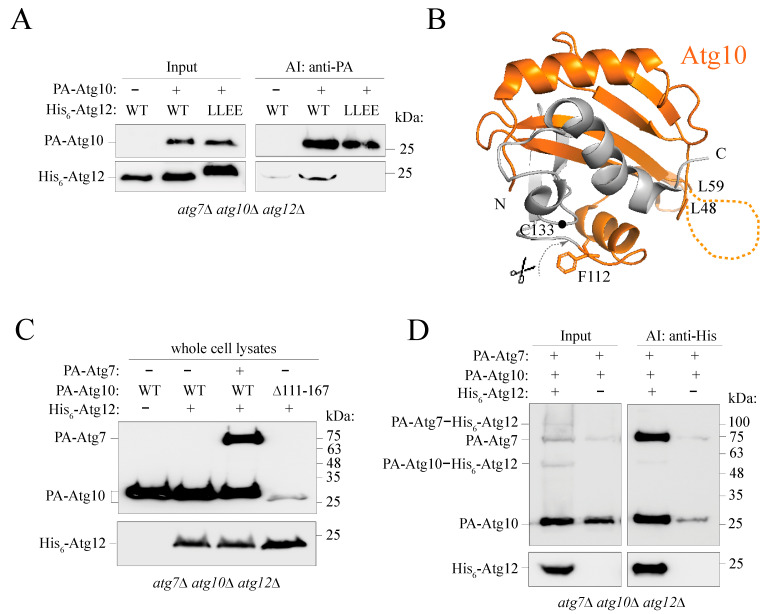
Probing the interaction between unconjugated Atg10 and Atg12. (**A**) Affinity-isolation experiment showing that unconjugated Atg10 and Atg12 interact in yeast *atg7*∆ *atg10*∆ *atg12*∆ cells in the absence of Atg7. The L54E L57E (LLEE) mutation in Atg12 abolishes this interaction. Atg10 is partially cleaved into a polypeptide of 13 kDa (27 kDa when fused to the PA tag of 14 kDa) that binds Atg12. (**B**) Crystal structure of Atg10 from yeast *S. cerevisiae* (PDB ID: 4EBR). Mapping the partial 13-kDa polypeptide of Atg10 on the crystal structure of full-length Atg10 from yeast shows that, in the absence of conjugation between catalytic C133 and Atg12 G186, the C terminus is cleaved off (gray) downstream of F112. The dark orange ribbon reveals partial Atg10 (13 kDa) that binds Atg12. The flexible region (dashed line) between L48 and L59 is invisible in the crystal structure. (**C**) Analysis of the protein levels in *atg7*∆ *atg10*∆ *atg12*∆ cell lysates shows that Atg10 is partially cleaved into the 13-kDa polypeptide independent of the presence or absence of Atg7 or Atg12 in yeast cells. The plasmid pCuHis_6_-Atg12(424) was transformed into cells overexpressing Atg10 variants on the plasmid pCuPA-Atg10(416) or pCuPA-Atg10[∆111-167](416). For the presence of Atg7, Atg7 and Atg10 were co-expressed on the dual-expression plasmid pCuPA-Atg7_CuPA-Atg10(416). Cells were cultured in nutrient-rich conditions to OD_600_~1. The lysates were TCA-precipitated, and the proteins were separated by SDS-PAGE and detected with anti-PA or anti-His antibodies. (**D**). Affinity isolation from *atg7*∆ *atg10*∆ *atg12*∆ yeast cells overexpressing PA-Atg7, PA-Atg10, and His_6_-Atg12 on the plasmids described in C. His_6_-Atg12 was pulled down by TALON metal affinity resin. Proteins were detected on western blots using anti-PA or anti-His antibodies.

**Figure 6 ijms-24-15036-f006:**
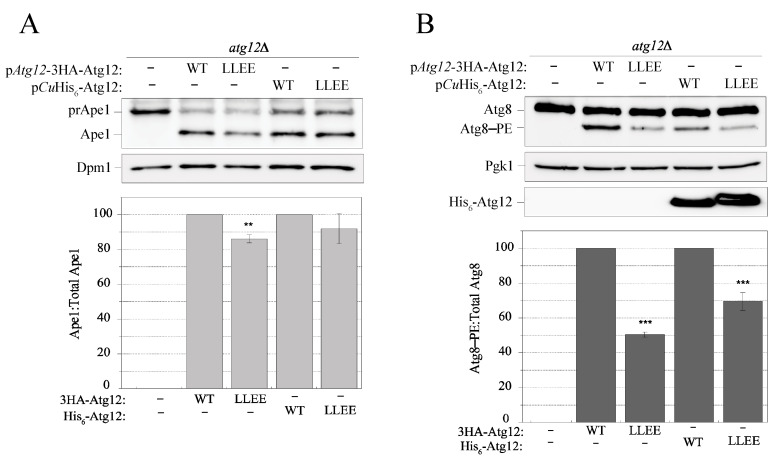
L54E L57E mutation decreases efficiency of the Cvt pathway and Atg8–PE conjugation. (**A**) The prApe1 processing assay as a readout of the Cvt pathway in nutrient-rich conditions. The wild type and L54E L57E (LLEE) variants were expressed with a 3HA tag under the control of the native *ATG12* promoter on the centromeric pRS416 plasmid or with the His_6_ tag under the control of the *CUP1* promoter on the pRS424 plasmid in *atg12∆* cells. Quantification of the Ape1:total Ape1 ratio was determined from three independent experiments. (**B**) Atg8 lipidation assessed in *atg12∆* cells expressing the Atg12 variants as in A. Cells were shifted to nitrogen-starvation medium for 4 h. The Atg8–PE:total Atg8 ratio was determined from three independent experiments. Error bars in panels A and B represent standard deviations. Statistical significance was tested using unpaired two-tailed Student’s *t* test: ** *p* < 0.005; *** *p* < 0.0005.

**Figure 7 ijms-24-15036-f007:**
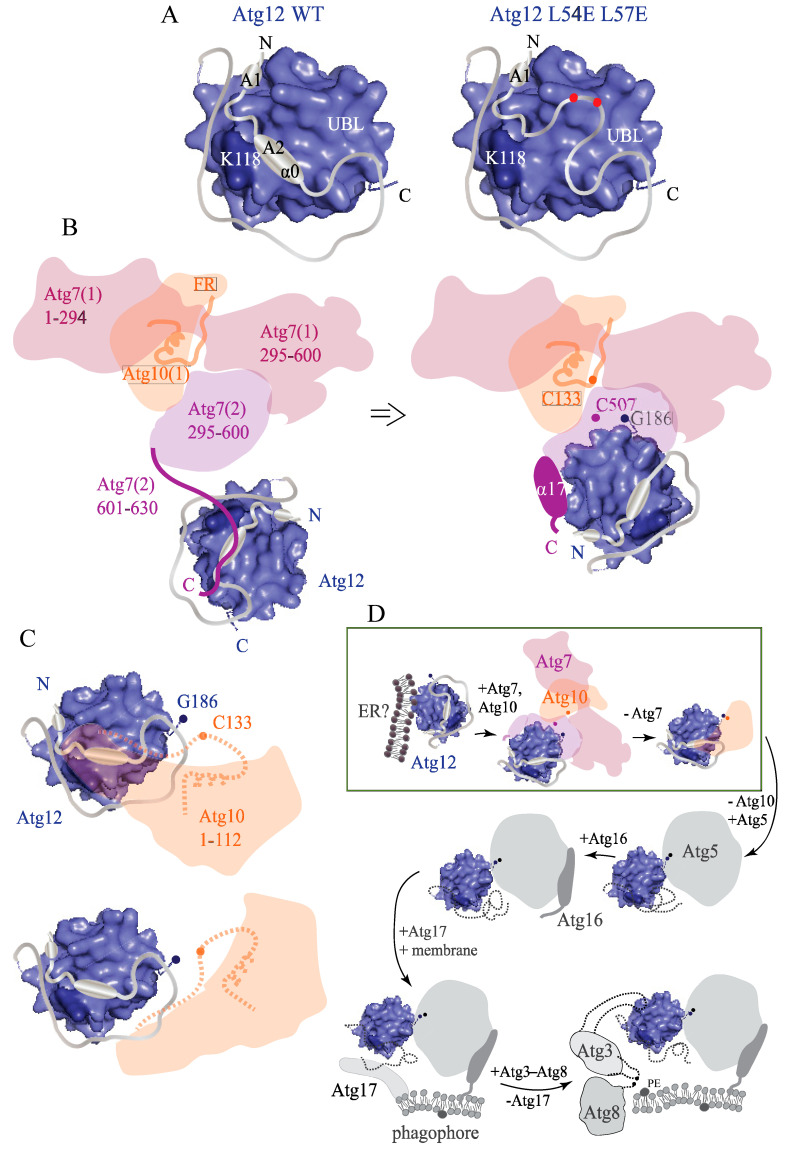
Putative models of the Atg12 protein from yeast. (**A**) In conjugation-free Atg12 wild-type protein (left), intramolecular interactions mediate packing of the N-terminal disordered region on the UBL domain. Conformational ensemble of the Atg12 IDPR brings K91, K93, K96, or K99 into close proximity to K118. The ANCHOR1 and ANCHOR2 domains gain stability via binding to the globular surface of the UBL domain, as the IDPR is too dynamic to fold and stabilize on its own. This binding presumably tightens folding of ANCHOR2 into a predicted α0 helix, which shields L54 and L57 from the solvent. A local disruption of the Atg12 α0 helix in the L54E L57E mutant (right) causes E54 and E57 to become exposed on the protein surface (red dots). An IDPR segment downstream of ANCHOR2 protects the C terminus from proteolytic cleavage in both wild type and L54E L57E Atg12. (**B**) Atg12 interacts with Atg7 in a manner similar to the Atg7–Atg8 interaction. The C-terminal disordered tail of Atg7 and the α0 helix in the IDPR of Atg12 are both essential for Atg12 binding to Atg7 (left). The C-terminal tail of Atg7 (601-630) partially folds into the α17 helix (601-613) for the Atg12 UBL domain to make contacts with the Atg7 activation domain (AD). Shielding of the Atg12 C terminus by the IDPR is released, and then Atg12 G186 is inserted into the Atg7 AD for conjugation with C507 (right). Note that Atg7 is present as a dimer and the Atg7(2) N-terminal domain (1-294) and the second Atg10, Atg10(2), are hidden behind the Atg7(1) 295-600 domain, which makes them invisible in this schematic representation of the Atg7–Atg10 heterotetramer. (**C**) Without conjugation of Atg10 C133 to Atg12 G186, the C-terminal region of Atg10 (downstream of F112) is proteolytically cleaved off (dashed orange line), yielding partial Atg10 that is able to noncovalently interact with Atg12 in an Atg12 IDPR-dependent manner. The Atg12 α0 helix either directly creates a binding site for partial Atg10 (upper) or ensures an optimal binding conformation of the Atg12 IDPR (lower). FR, flexible region. (**D**) Schematic depiction of the Atg12 lifetime when the protein engages in many interactions with binding partners. Unconjugated Atg12 adopts a compact conformation where the IDPR shields the C terminus until it forms the thioester conjugate with Atg7 and then Atg10. The Atg12 IDPR is required for efficient binding to both of these proteins. Atg10 mediates the linkage of G186 in Atg12 to Lys149 in Atg5. The Atg12-free surface of Atg5 in the Atg12–Atg5 conjugate binds noncovalently to Atg16. Atg17 recruits the Atg12–Atg5-Atg16 complex to the PAS on the phagophore membrane. Atg3 binds Atg12 via a large flexible loop in order to facilitate lipidation of Atg8. The green outlined rectangle highlights the portion of the Atg12 lifetime probed in this study. Conformations of the Atg12 IDPR after conjugation to Atg5 (dashed black line) remain undiscovered.

## Data Availability

Not applicable.

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
