# Peer review of "The Intrinsically Disordered N Terminus in Atg12 from Yeast Is Necessary for the Functional Structure of the Protein"

_ijms, 2023, doi:10.3390/ijms242015036_

Round 1

Reviewer 1 Report

Popelka et al shows the importance of intrinsically disordered protein region for the function of protein in ubiquitin-like conjugation cascade. Of all the proteins in the yeast ubiquitin-like conjugation cascade, Agt12 is the only protein that is not biochemically and structurally visualized. Through Alphafold, a deep learning based predictor, it was found that the N terminus domain of the Agt12 protein was a low confidence structure region. They have also shown that N terminus IDPR is needed for the functional conformation of the protein for its interaction with Agt7 or Agt10 along with the protection of the C terminal domain from proteolysis. The paper is well written and needs few minor corrections for acceptance to publication.  

  1. While the research has focussed on functionally characterized the Agt12-Agt7/Agt10 interaction, was the Agt12/Agt5/Agt16 affinity interaction characterized? If so, please provide details. 

  2. Were there any functional mutations identified in the IDPR region responsible for the interaction with Agt10 or Agt7? 

  3. It would help the background with a figure explaining the Agt12 interactions and what was confirmed in this paper as a role for the N terminus IDPR region.

Author Response

Reviewer #1

Popelka et al shows the importance of intrinsically disordered protein region for the function of protein in ubiquitin-like conjugation cascade. Of all the proteins in the yeast ubiquitin-like conjugation cascade, Agt12 is the only protein that is not biochemically and structurally visualized. Through Alphafold, a deep learning based predictor, it was found that the N terminus domain of the Agt12 protein was a low confidence structure region. They have also shown that N terminus IDPR is needed for the functional conformation of the protein for its interaction with Agt7 or Agt10 along with the protection of the C terminal domain from proteolysis. The paper is well written and needs few minor corrections for acceptance to publication.  

  1. While the research has focussed on functionally characterized the Agt12-Agt7/Agt10 interaction, was the Agt12/Agt5/Agt16 affinity interaction characterized? If so, please provide details. 

Interactions of the subunits within the Atg12–Atg5-Atg16 complex were not characterized in our study because we focused on unconjugated Atg12 and the interactions prior to the conjugation of Atg12 to Atg5. The life time of Atg12 is complex (see new Fig. 7D) and involves many protein-protein interactions, which cannot be addressed in one study.

  1. Were there any functional mutations identified in the IDPR region responsible for the interaction with Agt10 or Agt7? 

We are not aware of any previous study that tested functional mutations responsible for the interaction of Atg12 with Atg7 or Atg10. The L54E L57E in the IDPR of Atg12 presented in our study is, to our knowledge, the first mutation found that affects the interactions of Atg12 with the E2 and E3 enzymes (Fig. 4 and 5) and causes protein dysfunction (Fig. 6).

  1. It would help the background with a figure explaining the Agt12 interactions and what was confirmed in this paper as a role for the N terminus IDPR region.

We agree with the reviewer and have now included a new schematic figure (Fig. 7D) showing a complex involvement of Atg12 in many interactions during the protein’s lifetime. Findings presented in our study are highlighted by a green rectangle. Reference to figure 7D is in the “Conclusions” section.

Reviewer 2 Report

The ATG12-ATG5 covalent conjugation is crucial for the formation of autophagosome, which is a double-membrane vesicle that engulfs cellular components targeted for degradation. This ATG12 conjugation is an essential step in the autophagy pathway that contributes to the formation of autophagosomes and enables the degradation of cellular components. Furthermore, the E1-like enzyme ATG7 and E2-like enzyme ATG10 involve in conjugating the ubiquitin-like protein, ATG12, to its substrate, ATG5. Next, together with the ATG16 dimer, it forms the ATG12–ATG5-ATG16 complex. The ATG12–ATG5-ATG16 complex functions as an E3-like enzyme that facilitates ATG8 conjugation to phosphatidylethanolamine (PE) in the autophagic membranes. In the case of ATG12, the N-terminal disordered region might be involved in its interactions with other proteins in the autophagy pathway. This region could be responsible for mediating the conjugation of ATG12 to ATG5, which is a critical step in autophagosome formation. The flexibility of the disordered region could enable ATG12 to adopt different conformations to facilitate its interactions with other molecules. However, the specific functional roles and interactions of the ATG12 N-terminal disordered region and its contribution to the autophagy process, are still elusive.

Author here demonstrated that Intrinsic disordered protein region (IDPR) of Atg12 is crucial for the functional structure of the unconjugated protein, also IDPR of the protein is indispensable for Atg7 and Atg10 interaction. Overall, I think the findings presented in this manuscript are important and will be of interest to the broad readership of IJMS. In the manuscript many of the claims in the paper are well-justified. However, some important claims are not as well-substantiated by the provided data. In the following points, I highlight parts of the manuscript I found to be less compelling, and I offer some suggestions for how the authors might be able to fortify these elements of the paper.

Major concerns:

1) In supplementary figure 1A, authors should elaborate on the Multiple amino acid sequence alignment. For example, how and what parameters used for the alignment, sequence retrieval database and program used.

2) In the section 3.2; certain rational behind the experimental design is not clear. Authors should try to write in a more descriptive manner, that might help the readers from different field to understand better. Need more description, like for example, rational behind targeting only specific conserved residues in ANCHOR 1 and 2, as some other corresponding residues in the region are also conserved, ex. Ser(S) in ANCHOR 1 and  Glu(E) in ANCHOR 2. Also, a graph for prApe1:total Ape1 based on Figure 1B densitometry could be useful.

3) In section 3.3; generally conjugation requires the C-terminal glycine in ATG8. While ATG12 ends in glycine. Typically ATG8 needs C-terminal processing by the cysteine protease ATG4 to expose it. ATG4 also deconjugates ATG8 from the membrane for recycling purposes. Authors must verify the result (presented in figure S1B, C, the impaired membrane localization of Atg12-AT8 chimeric protein) by expressing ATG4.

4) In section 3.5; the data presented is interesting. However, authors did not mention about the plausible factor regulating the proteolysis of ATG12. Also, the discussion on the physiological relevance of this proteolytic cleavage could add a value to this data presented in Figure 3.

5) In Figure 3, the proteolytic cleavage of N/C-terminal, could be confirmed additionally by an His/Flag-pulldown experiment. Authors must check whether anti-His anti-flag could pulldown His6-Atg12-FLAG and His6-Atg12(Δ3-79)-FLAG. Also authors should estimate the percentage of proteolytic fraction and presented in a quantitative form. According to Figure B, certain fractions of Atg12Δ3-79, did not undergo C-terminal proteolysis. Author should clearly explain about the reason of such partial proteolysis phenomenon observed.

Minor concerns:

1) All the figures should be consistent and high quality. Why authors changed the marker here in Figure 3E and 3F, as improper resolution of the gel might mask off detectable bands.

2) The nomenclature of the ATg12 mutant is bit confusing. Rather than writing His6-Atg12-FLAG Δ3-79, writing His6-Atg12Δ3-79(superscript)-FLAG  could be more appropriate.

3) Line 55-56 : “The Atg12–Atg5 conjugate interacts with Atg17 to target the E3-like enzyme to the phagophore assembly site (PAS)”. Author should correct it. It’s not Atg17, should be Atg16.

Author Response

Reviewer #2

The ATG12-ATG5 covalent conjugation is crucial for the formation of autophagosome, which is a double-membrane vesicle that engulfs cellular components targeted for degradation. This ATG12 conjugation is an essential step in the autophagy pathway that contributes to the formation of autophagosomes and enables the degradation of cellular components. Furthermore, the E1-like enzyme ATG7 and E2-like enzyme ATG10 involve in conjugating the ubiquitin-like protein, ATG12, to its substrate, ATG5. Next, together with the ATG16 dimer, it forms the ATG12–ATG5-ATG16 complex. The ATG12–ATG5-ATG16 complex functions as an E3-like enzyme that facilitates ATG8 conjugation to phosphatidylethanolamine (PE) in the autophagic membranes. In the case of ATG12, the N-terminal disordered region might be involved in its interactions with other proteins in the autophagy pathway. This region could be responsible for mediating the conjugation of ATG12 to ATG5, which is a critical step in autophagosome formation. The flexibility of the disordered region could enable ATG12 to adopt different conformations to facilitate its interactions with other molecules. However, the specific functional roles and interactions of the ATG12 N-terminal disordered region and its contribution to the autophagy process, are still elusive.

Author here demonstrated that Intrinsic disordered protein region (IDPR) of Atg12 is crucial for the functional structure of the unconjugated protein, also IDPR of the protein is indispensable for Atg7 and Atg10 interaction. Overall, I think the findings presented in this manuscript are important and will be of interest to the broad readership of IJMS. In the manuscript many of the claims in the paper are well-justified. However, some important claims are not as well-substantiated by the provided data. In the following points, I highlight parts of the manuscript I found to be less compelling, and I offer some suggestions for how the authors might be able to fortify these elements of the paper.

Major concerns:

1) In supplementary figure 1A, authors should elaborate on the Multiple amino acid sequence alignment. For example, how and what parameters used for the alignment, sequence retrieval database and program used.

We agree with the reviewer that information on multiple amino acid sequence alignment was missing. We have now added this necessary information to section 2.8. Bioinformatics analysis. New text (in bold) now reads: “Amino acid sequences for multiple sequence alignments were obtained from the UniProt database. The alignment was created using the ClustalW Multiple alignment algorithm in the BioEdit program [54].”

2) In the section 3.2; certain rational behind the experimental design is not clear. Authors should try to write in a more descriptive manner, that might help the readers from different field to understand better. Need more description, like for example, rational behind targeting only specific conserved residues in ANCHOR 1 and 2, as some other corresponding residues in the region are also conserved, ex. Ser(S) in ANCHOR 1 and  Glu(E) in ANCHOR 2. Also, a graph for prApe1:total Ape1 based on Figure 1B densitometry could be useful.

We appreciate this reviewer’s comment. We now added in line 207 (line in the MDPI format of the original manuscript) our rationale for selection of mutations. The new text now reads: “…(Fig. 1A, S1A), to disrupt hydrophobic and/or electrostatic interactions mediated by these elements. Mutagenesis of conserved serine residues was omitted due to the lack of information on regulation of Atg12 via phosphorylation.

Figure 1B now includes a new panel under the western blots that shows a graph for the Ape1:Total Ape1 ratio evaluated by densitometry from three independent experiments.

3) In section 3.3; generally conjugation requires the C-terminal glycine in ATG8. While ATG12 ends in glycine. Typically ATG8 needs C-terminal processing by the cysteine protease ATG4 to expose it. ATG4 also deconjugates ATG8 from the membrane for recycling purposes. Authors must verify the result (presented in figure S1B, C, the impaired membrane localization of Atg12-AT8 chimeric protein) by expressing ATG4.

We understand the reviewer’s concern; however, we respectfully disagree with this comment. The subcellular fractionation experiments were intentionally carried out in the multiple knockout (MKO) cells lacking most of the autophagy proteins, including Atg4. This approach eliminated a possible membrane-binding of Atg12 or Atg8 via the autophagy machinery. The Atg4 protease has a membrane binding capability (as illustrated in the figure below this paragraph; our unpublished data: M, membrane fraction; C, cytosolic fraction) and has been proposed to carry a membrane-binding loop [Maruyama & Noda, The Journal of Antibiotics (2018) 71; 72-78]. Therefore, the presence of Atg4 along with Atg8 or the Atg12N-Atg8 chimeric protein in MKO cells would interfere with interpretation of membrane binding in subcellular fractionation experiments. Please note that Atg8 in our experiments carries the C-terminal arginine because we used nascent Atg8, before it enters the conjugation cascade, solely as a ubiquitin-like probe to test the Atg12 IDPR. Furthermore, Atg8 modifies its N terminus that transitions from a disordered to helical conformation after processing of the C-terminal arginine (Schwarten et al., Biochem. Biophys. Res. Commun. (2010) 395, 426-431], which complicates interpretation of results in the presence of Atg4 because the Atg12 IDPR is attached to the Atg8 N terminus in the chimeric protein. A thorough structural study would be required to elucidate the effect of this conformational change on the N terminus of the processed Atg12N-Atg8 chimera, a problem that is clearly out of the scope of this study. Finally, Atg8 processing by Atg4 in MKO cells expressing both proteins under the control of the CUP1 promoter has not been studied and roles of the promoter (CUP1 vs. native) as well as the rest of the missing autophagy machinery in this process are not explored.

Please, see the attachment for the figure

4) In section 3.5; the data presented is interesting. However, authors did not mention about the plausible factor regulating the proteolysis of ATG12. Also, the discussion on the physiological relevance of this proteolytic cleavage could add a value to this data presented in Figure 3.

We think this is a valid comment. To address this reviewer’s point, we have now added a sentence in line 313 (line in the MDPI format of the original manuscript). The new text now reads:” …the unconjugated C terminus. Regulation of such a temporary shielding, until the C-terminal tail in Atg12 enters the hydrophobic cavity of Atg7, remains elusive. Nevertheless, our data indicate that plasticity of the N-terminal IDPR is an important physiological determinant of the Atg12 structure.”

5) In Figure 3, the proteolytic cleavage of N/C-terminal, could be confirmed additionally by an His/Flag-pulldown experiment. Authors must check whether anti-His anti-flag could pulldown His6-Atg12-FLAG and His6-Atg12(Δ3-79)-FLAG. Also authors should estimate the percentage of proteolytic fraction and presented in a quantitative form. According to Figure B, certain fractions of Atg12Δ3-79, did not undergo C-terminal proteolysis. Author should clearly explain about the reason of such partial proteolysis phenomenon observed.

We thank the reviewer for these comments as they help to improve our manuscript. We appreciate the reviewer’s suggestion to try a pulldown with the His tag.  We carried out this experiment using TALON metal affinity resin. The result is now in a new panel of figure 3. As shown in this figure, the affinity isolation experiment (Fig. 3D) mirrors the data from whole cell lysates (Fig. 3A,B). A population of intact ∆3-79 is detected with anti-FLAG antibody and the multiple-isoform pattern for ∆3-79 is detected with anti-His antibody in both the lysates (Input) and pulldowns (AI: anti-His). This result confirms that ∆3-79 has isoforms of various lengths and that the bands detected by anti-His antibody on western blots are not nonspecific. A second repeat of the His-tag pulldown showed a similar result, as illustrated in the figure below this paragraph. In light of these data, we consider anti-FLAG pulldowns unnecessary, as they would be redundant.

Please, see the attachment for the figure

The reviewer raised a valid point regarding the percentage of the proteolytic fraction. We now included a new panel in Fig. 3C with the quantitative evaluation of the percentage of major proteolytic isoform relative to the total protein detected by western blot. The new text now reads: “…with anti-His antibody shows that a significant population of ∆3-79 had the C terminus cleaved off (Fig. 3A-C). Affinity isolation using the His tag confirmed that ∆3-79 had isoforms of various lengths (Fig. 3D). This result shows…”.

We also included a sentence in line 308 to explain what we think is a reason for a subpopulation of ∆3-79 detected with the intact C terminus. The new text now reads: “Taking into account the structural plasticity of IDPRs, a conformational ensemble of the truncated Atg12 IDPR swinging near the C terminus in this subpopulation can be seen as a cloud occluding the access of proteases”.

Minor concerns:

1) All the figures should be consistent and high quality. Why authors changed the marker here in Figure 3E and 3F, as improper resolution of the gel might mask off detectable bands.

In the original Figure 3E and 3F, we used a marker that shows the molecular masses of 20 and 17 kDa to determine more accurately the position of the ∆3-27 mutant on the western blot. We agree that figures should be consistent as regards resolution. We have now matched the resolution in panels A, B, D, G, and H, in a new figure 3, where panels G and H reveal that there are no detectable bands below 17 kDa on western blots.

2) The nomenclature of the ATg12 mutant is bit confusing. Rather than writing His6-Atg12-FLAG Δ3-79, writing His6-Atg12Δ3-79(superscript)-FLAG could be more appropriate.

In general, superscript notations are used to denote point mutants. Therefore, we think it is less confusing for the readers to use the bracketed notation for deletions.

3) Line 55-56: “The Atg12–Atg5 conjugate interacts with Atg17 to target the E3-like enzyme to the phagophore assembly site (PAS)”. Author should correct it. It’s not Atg17, should be Atg16.

The sentence on the line 55-56 is correct. Harada et al. showed that the Atg12–Atg5 conjugate binds Atg17 (ref. 8). We appreciate this comment, which reveals that the original sentence was confusing. We now reworded the sentence to read: “The Atg12–Atg5-Atg16 complex interacts with Atg17 to target the E3-like enzyme to the phagophore assembly site (PAS)”.

Reviewer 3 Report

Popelka et al. performed biochemical analysis on components involved in the process of macroautophagy, in particular, the ATG12 protein and its activators ATG7 and ATG10. The authors focused on the intrinsically disordered sequence of ATG12 and found that it is critical for its function. The ATG12 protein is composed of the intrinsically disordered protein region (IDPR) at its N terminus and a UBL domain at its C terminus. The IDPR, while unstructured, contains two evolutionarily conserved segments which overlaps with the two predicted ANCHOR sequences. The authors discovered that ANCHOR2 within the IDPR is critical for mediating nonselective autophagy, but IDPR itself does not confer association with cell membrane. Crosslinking and mass spec analysis revealed that the IDPR is positioned close to the UBL domain. Further, the IDPR mediated intramolecular interaction is important in shielding the UBL domain from protease cleavage and also in protecting the N terminus from being cleaved. Lastly, the authors demonstrated the critical role of the IDPR in mediating intermolecular interactions of ATG12 with ATG7 and ATG10: alpha0 helix required for binding ATG7 and essential for noncovalent Atg10-Atg12 interface. They also discovered that when ATG10 is free and not conjugated with ATG12, it is prone to proteolysis at the C terminus.

This study was well done with carefully designed experiments and the manuscript was clearly and logically written. This manuscript provides mechanistic insight into the function of the intrinsically disordered sequence of ATG12, the understanding of which will advance our knowledge in diseases  caused by mutations in the similar region of the human ATG12. This manuscript will be of interest to the general audience at IJMS.

I have a few minor comments listed below:

1.       Line 447-450 and figure 5D. It is not clear to me how this assay was performed? Did you use anti-HIS resin to pull down His-Atg12 and then detected proteins using anti-PA antibodies? Why was the 56KD PA-Atg10-His-atg12 conjugate only visible on the left “input” but not on the right part?

2.       Line 55: typo? “Atg16”

3.       Line 323-324: could you add a sentence in between, because the transition to proteolysis of the N terminus is very abrupt, as the preceding sentences were still talking about the cleavage at the C terminus, and also the title of this section is also on “…protect the unconjugated UBL domain from proteolysis” which is on the C terminus

4.       Figure 4: please align the “+” and alphabets on the next line on the right part of figure4A.

5.       Line 489: add “3.8” in front of the title.

Author Response

Reviewer # 3

Popelka et al. performed biochemical analysis on components involved in the process of macroautophagy, in particular, the ATG12 protein and its activators ATG7 and ATG10. The authors focused on the intrinsically disordered sequence of ATG12 and found that it is critical for its function. The ATG12 protein is composed of the intrinsically disordered protein region (IDPR) at its N terminus and a UBL domain at its C terminus. The IDPR, while unstructured, contains two evolutionarily conserved segments which overlaps with the two predicted ANCHOR sequences. The authors discovered that ANCHOR2 within the IDPR is critical for mediating nonselective autophagy, but IDPR itself does not confer association with cell membrane. Crosslinking and mass spec analysis revealed that the IDPR is positioned close to the UBL domain. Further, the IDPR mediated intramolecular interaction is important in shielding the UBL domain from protease cleavage and also in protecting the N terminus from being cleaved. Lastly, the authors demonstrated the critical role of the IDPR in mediating intermolecular interactions of ATG12 with ATG7 and ATG10: alpha0 helix required for binding ATG7 and essential for noncovalent Atg10-Atg12 interface. They also discovered that when ATG10 is free and not conjugated with ATG12, it is prone to proteolysis at the C terminus.

This study was well done with carefully designed experiments and the manuscript was clearly and logically written. This manuscript provides mechanistic insight into the function of the intrinsically disordered sequence of ATG12, the understanding of which will advance our knowledge in diseases  caused by mutations in the similar region of the human ATG12. This manuscript will be of interest to the general audience at IJMS.

I have a few minor comments listed below:

  1. Line 447-450 and figure 5D. It is not clear to me how this assay was performed? Did you use anti-HIS resin to pull down His-Atg12 and then detected proteins using anti-PA antibodies? Why was the 56KD PA-Atg10-His-atg12 conjugate only visible on the left “input” but not on the right part?

      Yes, the reviewer is correct. We used TALON metal affinity resin to pulldown His6-Atg12 and detect the interacting proteins using anti-PA antibody. The Figure 5 legend describing panel D provides this information.

      We appreciate this comment. After a closer look, the PA-Atg10–His6-Atg12 conjugate (~56 kDa) is detectable also in the AI sample, although at a low intensity, as illustrated in the figure below this paragraph. For a reason unknown to us, the unconjugated protein complexes involving PA-Atg7, PA-Atg10, and His6-Atg12 appear to outcompete on the resin the protein complexes containing the thioester conjugates (PA-Atg7–His6-Atg12 and PA-Atg10–His6-Atg12). A reason could be steric effects. We corrected the text in line 449 by removing “in the input”.                                             

         Please see the attachment for the figure         

  1. Line 55: typo? “Atg16”

As in the response to Reviewer #2 who brought up the same concern, the sentence in line 55-56 is correct. Harada et al. showed that the Atg12–Atg5 conjugate binds Atg17 (ref. 8). We appreciate this comment, which reveals that the original sentence was confusing. We now reworded the sentence to read: “The Atg12–Atg5-Atg16 complex interacts with Atg17 to target the E3-like enzyme to the phagophore assembly site (PAS)”.

  1. Line 323-324: could you add a sentence in between, because the transition to proteolysis of the N terminus is very abrupt, as the preceding sentences were still talking about the cleavage at the C terminus, and also the title of this section is also on “…protect the unconjugated UBL domain from proteolysis” which is on the C terminus

      We appreciate this reviewer’s comment as it helps us to improve the readability of the manuscript. We now added a new text in line 319 to read: “… binding to a globular protein [53]. In our next experiments we focused on a part of the Atg12 N terminus carrying ANCHOR1. Given the dispensability of ANCHOR1 for protein function in nonselective autophagy (Fig. 1B), we asked whether ANCHOR1 has a structural contribution in the N terminus of unconjugated Atg12”.

The reviewer has a valid point, the title of section 3.5 did not include all data presented in Figure 3. We have now modified the title of this section. The new title now reads: “Intramolecular interactions between the IDPR and UBL secure the intact structure of unconjugated Atg12”.

  1. Figure 4: please align the “+” and alphabets on the next line on the right part of figure4A.

      Done. We thank the reviewer for noticing this misalignment.

  1. Line 489: add “3.8” in front of the title.

We appreciate that the reviewer noticed this formatting error. It happened during transition of the manuscript into the MDPI format. We will inform the publisher and watch for the correction in the proofs.

Round 2

Reviewer 2 Report

The revised manuscript is improved and addressed all the necessary concerns in a point wise manner. The authors are applauded for putting effort in making the manuscript more structured.